# Batik Nitik 960 Dataset for Classification, Retrieval, and Generator

Agus Eko Minarno [1,2,*], Indah Soesanti [1] and Hanung Adi Nugroho [1,*]

1 Department of Electrical and Information Technology, Jl. Grafika 2, Universitas Gadjah Mada, Yogyakarta 55281, Indonesia
2 Department of Information Technology, Jl. Raya Tlogomas 246, Universitas Muhammadiyah Malang, Malang 65144, Indonesia
* Correspondence: aguseko@umm.ac.id (A.E.M.); adinugroho@ugm.ac.id (H.A.N.)

**Abstract:** Batik is one of the traditional heritages of Indonesia, with each motif of batik having a profound cultural and philosophical significance. This article introduces Batik Nitik 960 dataset from Yogyakarta, Indonesia. The dataset was extracted from a piece of fabric with 60 Nitik patterns. The dataset was supplied by the Paguyuban Pecinta Batik Indonesia (PPBI) Sekar Jagad Yogyakarta collection of Winotosasto Batik and the data were extracted from the APIPS Gallery. Each of the 60 categories in the collection contains 16 photographs, for a total of 960 images. The photographs were acquired with a Sony Alpha a6400, illuminated with a Godox SK II 400, and the data were compressed using the jpg file format. Each category contains four motifs rotated by 90, 180, and 270 degrees. Thus, the total number of images per motif is 16. Each class has a specific philosophical significance associated with the motif's origins. This dataset aims to enable the training and evaluation of machine learning models for classification, retrieval, or generation of a new batik pattern using a generative adversarial network. To our knowledge, this study is the first to present a Batik Nitik dataset equipped with philosophical significance that is freely accessible.

**Dataset:** http://doi.org/10.17632/sgh484jxzy.3

**Dataset License:** : CC BY 4.0

**Keywords:** batik; nitik; dataset; deep learning; classification; image retrieval; generative adversarial network

## 1. Summary

Batik Nitik is a highly valued form of textile art, originating in Indonesia and spreading throughout the world, characterized by its unique patterns and motifs, resulting from a long history of cultural exchange and acculturation. Despite its diversity, the study of batik experienced several challenges, including the difficulty in identifying, classifying, and searching for specific motifs and patterns. Elaborating further, the lack of innovation in the development of new motifs and patterns also presents a problem, leading to a sense of stagnation in the field of research. As such, to address these challenges, the present researchers have proposed various methods, including Content Based Image Retrieval (CBIR) [1–22], classification techniques [23–42], and the implementation of generative models [43–48].

In general, CBIR methods play a role in searching for relevant batik patterns based on their visual content, thereby assisting researchers and practitioners in the identification and retrieval of specific patterns. Furthermore, classification techniques are beneficial in grouping similar patterns together, providing a more comprehensive understanding toward the diversity of batik patterns.

Previously, other generative models, such as Generative Adversarial Networks (GANs), have also been implemented to develop new batik patterns, addressing the problem of stagnation in the field of research. GANs have potential in generating new patterns similar to

existing ones while also introducing unique and innovative elements. This study proposes various methods to sustain the field of batik dynamics, leading to new insights into the study of batik.

However, the study of batik is characterized by several limitations. One of the main challenges is posed by the lack of well-annotated datasets which provide accurate information regarding the patterns, origin, and philosophical meaning. The Batik 300 dataset, published by Minarno [49], provides a large collection of batik images including adequate metadata to support meaningful analysis and interpretation. The Batik Nitik 960 dataset, on the other hand, becomes the first publicly available dataset providing rich metadata, comprising the name of the motif, its category, and its philosophical meaning [50].

Batik 300 is a general collection of 50 classes of batik motifs, while Batik 960 focuses on 60 classes of Nitik motifs; therefore, Batik 960 significantly differs from Batik 300. The Batik 300 dataset only utilizes labels, without a specific motif name, with no metadata in the form of motif philosophy, and it was never validated by experts. Meanwhile, Batik Nitik 960 provides a specific motif only focusing on the Nitik motif type which includes a motif name and philosophical meaning, validated by batik experts. Hence, Batik Nitik 960 represents the first complete primary dataset with metadata and can be accessed publicly. In addition, the metadata on Batik Nitik 960 are suitable for application as additional information in classification results, thereby allowing classification results to include the motif name and the philosophical meaning of the classified motif. Subsequently, the metadata of Batik Nitik 960 are also utilized as additional text-based queries along with image queries in the searching or classification of batik motifs.

## 2. Data Description

Batik Tulis Nitik is one of the oldest typical Yogyakarta motifs developed by members of the Yogyakarta Palace BRAy. Brongtodiningrat on 19 February 1940. The 56 Nitik motifs by BRAy. Brongtodiningrat were successively remastered by Haryani Winotosastro. Haryani added four motifs, resulting in a total of 60 motifs, including Sekar Jeruk, Sekar Srengenge, Sekar Sawo, and Sekar Gambir. Figure 1 illustrates a sample of the Nitik Batik fabric with the 60 motifs belonging to Haryani Winotosastro. The value of the dataset lies in the following:

- The data are used for computer vision and pattern recognition, employed in the generative adversarial network (GAN), classification, or image retrieval.
- Batik Nitik 960 dataset motivates researchers to develop innovative methods for generating new motifs using GAN.
- The data encourage researchers to classify or build retrieval models based on motifs and metadata using machine learning or deep learning.
- The Batik Nitik images are augmented and ready to be used in machine learning.

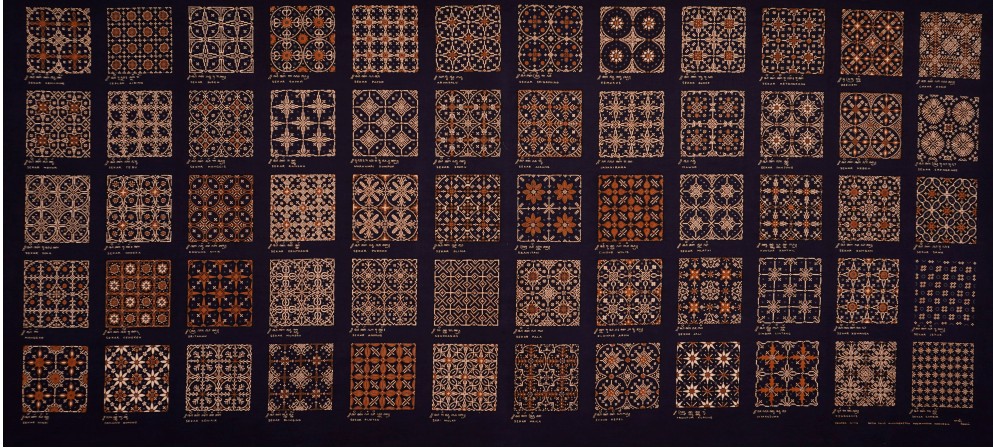

**Figure 1.** The remastered version (in 2010) of Batik Nitik fabric belonging to Winotosastro.

The dataset comprises 60 categories with a total of 960 images, having a dimension of 512 × 512 pixels in jpg format. The original 60 categories are depicted in Figure 2, including the name of the motif. Each category of a batik motifs presents a background and philosophical meaning. A description of the dataset is provided in Table 1.

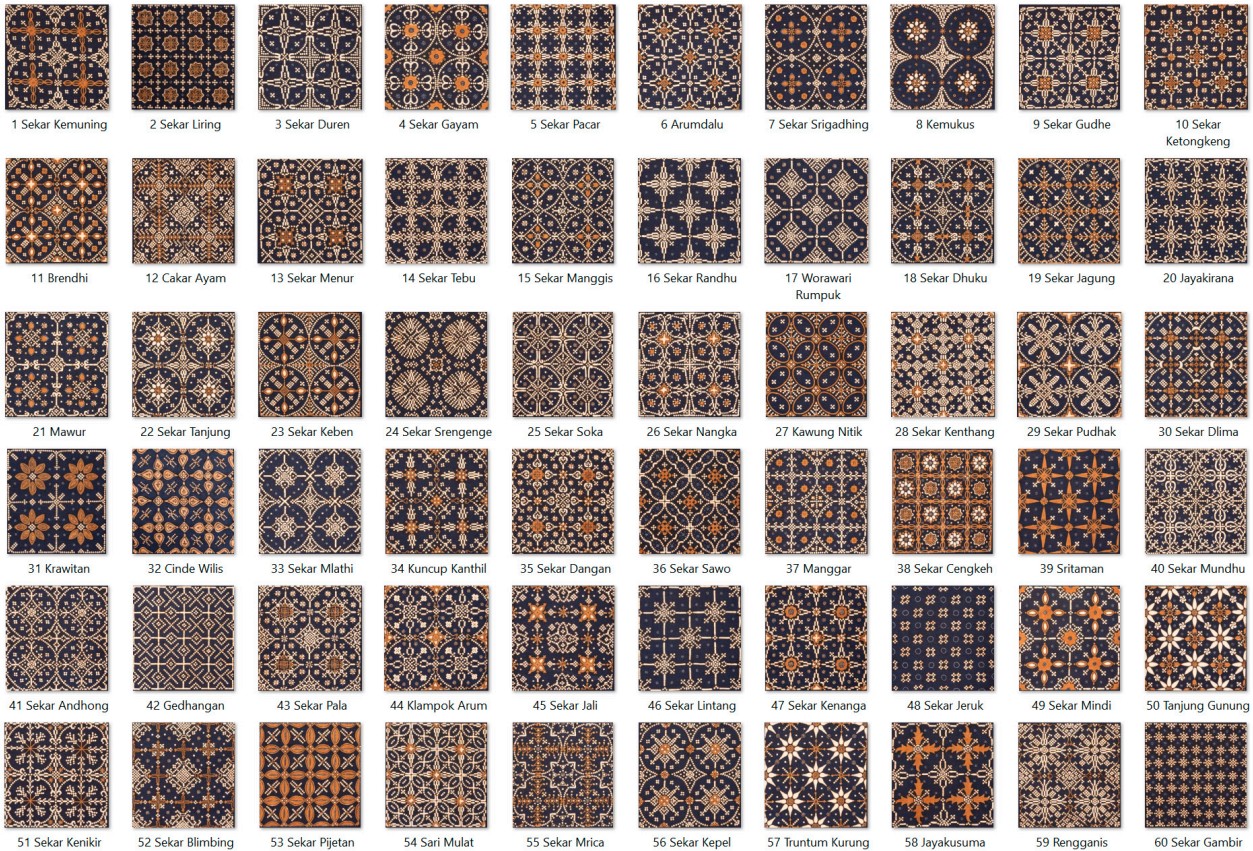

**Figure 2.** The 60 categories within Batik Nitik 960.

**Table 1.** Specification of the dataset.

| Subject | Computer Vision and Pattern Recognition |
|---|---|
| Specific subject area | Generative Adversarial Network, Classification, Image Retrieval. |
| Type of data | Image |
| How the data were acquired | Data acquisition was made using a Sony Alpha 6400 camera with sensor APS-C, resolution 24 MP, image dimension of 6024 × 4024 pixels, double lighting using Godox SKII400, Trigger Godox X2T for Sony, Sony Lens 80–135 mm, format Raw and RGB colored. |
| Data format | Filtered (.jpg) |
| Description of data collection | Data were provided by a collaboration of Paguyuban Pecinta Batik Indonesia (PPBI) Sekar Jagad Yogyakarta and Universitas Muhammadiyah Malang. Images were obtained from a piece of fabric that consists of a sixty-piece motif from the Winotosastro Batik collection. Each sample piece consisted of four motifs, and each motif was rotated by 90, 180, and 270 degrees. The total data consisted of 960 images and 60 categories, each comprising 16 images. The present researcher named this dataset "Batik Nitik 960" to support batik research. |
| Data source location | Institution: Paguyuban Pecinta Batik Indonesia (PPBI) Sekar Jagad<br>· City/Town/Region: Yogyakarta<br>· Country: Indonesia |

| Subject | Computer Vision and Pattern Recognition |
|---|---|
| Data accessibility | All the images were uploaded to an open, free-to-use research data repository entitled "Mendeley Data." The specific details required to access the data are as follows: Minarno, Agus Eko; Nugroho, Hanung Adi; Soesanti, Indah (2022), "Batik Nitik 960", Mendeley Data, V3, https://doi.org/10.17632/sgh484jxzy.3 (accessed on 23 January 2023). Repository name: Mendeley Data Data identification number: https://doi.org/10.17632/sgh484jxzy.3 (accessed on 23 January 2023). Direct URL to data: http://doi.org/10.17632/sgh484jxzy.3 (accessed on 23 January 2023). |

## 3. Materials and Methods

### 3.1. Data Acquisition

The images were captured in September 2022 in APIP's Batik, Yogyakarta, Indonesia. The studio size was 10 × 8 m. An image of a piece of Batik Nitik fabric was captured using a Sony Alpha 6400 camera with Sony lens 85–135 mm and lighting consisting of a two-set Godox II SK 400. The camera setting relied on features such as F1/10, a shutter speed of 1/10, ISO 200, a focal length of 135, a white balance of 5500 K, and a flashlight of 1/16. The image dimension was 6024 × 4024 pixels in raw format; the image was converted using jpg format. The Nitik motif was captured one by one to capture the details of the texture. Figure 3 illustrates the Batik-Nitik-fabric-capturing process. The preprocessing of images occurred in three phases: image cropping, image splitting, and image augmentation.

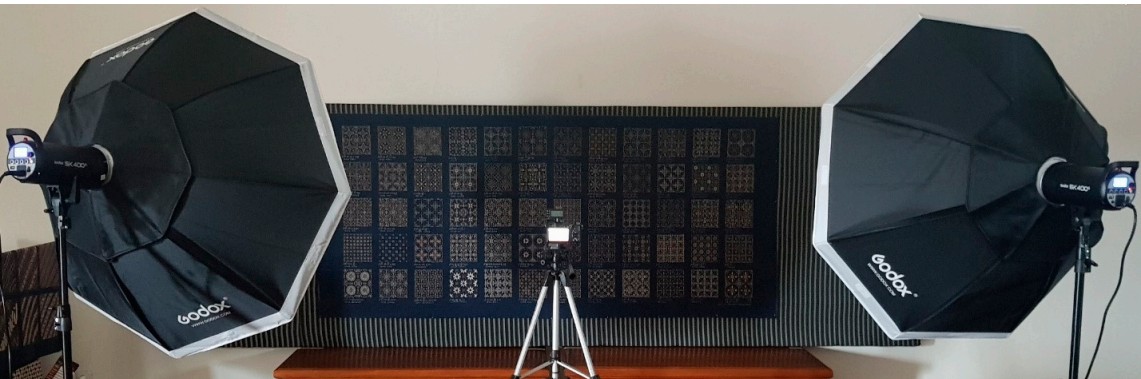

**Figure 3.** Process of capturing Batik Nitik fabric using a Sony A6400 camera and Godox II SK 400.

### 3.1.1. Image Cropping

A piece of the motif was captured in raw format and cut one by one manually, undergoing cropping process which required 60 iterations. The cropping process is illustrated in Figure 4.

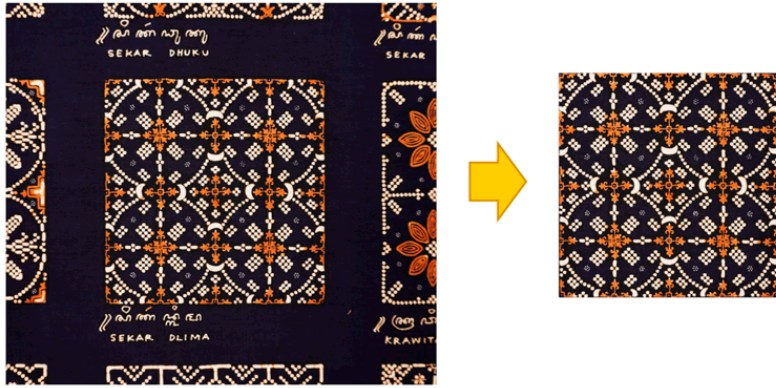

**Figure 4.** Preprocessing stage of cropping each sample piece of Batik Nitik motif.

### 3.1.2. Image Splitting

The second phase consisted of splitting the four sample motifs into four individual motifs. As a result of this process, the size of the dataset was 60 × 4 = 240 images. The splitting process is illustrated in Figure 5.

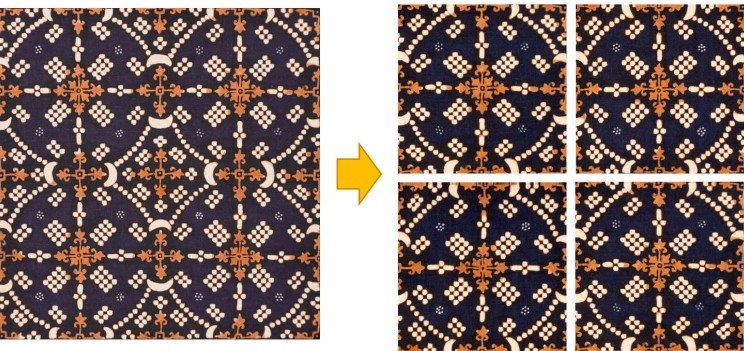

**Figure 5.** Preprocessing stage of splitting each sample piece of Batik Nitik motif into four separate pieces.

### 3.1.3. Image Augmentation

In this phase, the number of images was increased through rotation of the original motif image; each image was rotated by 90, 180, and 270 degrees, generating 240 × 4 = 960 images. The process of augmentation is illustrated in Figure 6.

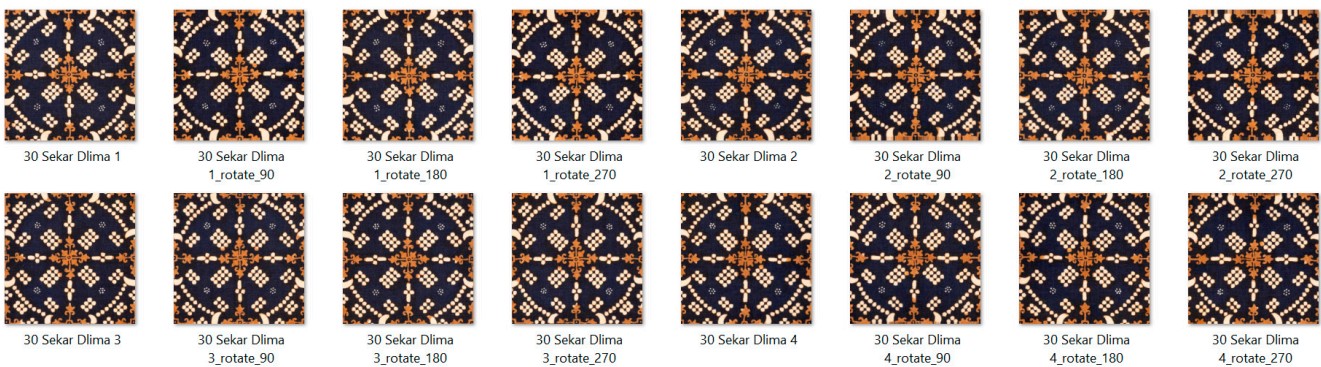

**Figure 6.** Preprocessing stage of augmentation of Batik Nitik motif by rotating the original motif image by 90, 180, 270 degrees.

### *3.2. Metadata*

Batik Nitik 960 consists of 60 motifs, with each motif associated with a philosophical meaning described by batik experts. Table 2 presents seven samples of the Batik Nitik motifs and their associated philosophical meaning. Full images and metadata can be accessed at the dataset link.

**Table 2.** Seven samples of Batik Nitik 960 metadata.

| No. | Batik Nitik Motif | Description |
|---|---|---|
| 1 | 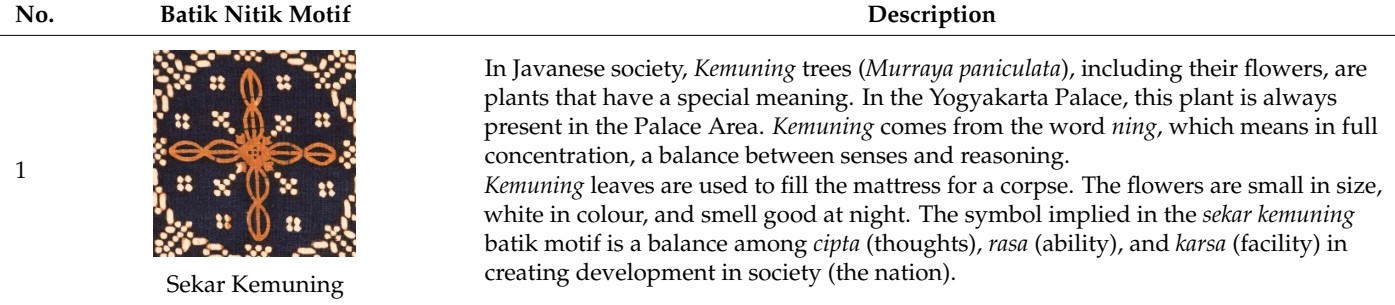 Sekar Kemuning | In Javanese society, *Kemuning* trees (*Murraya paniculata*), including their flowers, are plants that have a special meaning. In the Yogyakarta Palace, this plant is always present in the Palace Area. *Kemuning* comes from the word *ning*, which means in full concentration, a balance between senses and reasoning. *Kemuning* leaves are used to fill the mattress for a corpse. The flowers are small in size, white in colour, and smell good at night. The symbol implied in the *sekar kemuning* batik motif is a balance among *cipta* (thoughts), *rasa* (ability), and *karsa* (facility) in creating development in society (the nation). |

**Table 2.** *Cont.*

| No. | Batik Nitik Motif | Description |
|---|---|---|
| 2 | Ceplok Liring | *Ceplok* is a term in the world of batik referring to the irregular placement of motifs in a field. *Liring* in Javanese means looking at something from one corner of the eye (not fully). This motif conveys a normative meaning for humans, encouraging the latter not to take anything for granted, in terms of a person's behaviour, thoughts, and opinions. This is because there is a possibility that things that are considered trivial will have a big impact. Therefore, a caring attitude and respect for others must be set forth. |
| 3 | Sekar Duren | The durian flower (*Durio*) is called *dlongop* in Javanese, a term which refers to the human attitude which manifests itself in an atmosphere that exclude critical thinking. This warns us all that the *dlongop* attitude is not commendable for a creative community, as it does not support the emergence of ideas and innovations. |
| 4 | Sekar Gayam | *Gayam* in the Javanese society has the philosophical meaning of *gayuh*, which means achieving goals, and *ayem*, which means serene and peaceful. *Gayam* (*Inocarpus fagifer*) is the name of a tree that generally grows large and produces edible fruit which cause a feeling of fullness when eaten. In Yogyakarta, *gayam* trees were planted for shade during the Islamic Mataram kingdom. *Gayam* trees also grow well around natural springs, making the atmosphere under the *gayam* tree definitely refreshing. The *sekar gayam* batik motif is a symbol of shade that creates a sense of calm (peaceful). It also encourages the wearer to have a purpose in life that is calming both for themselves and their environment, just like the *gayam* tree. |
| 5 | Sekar Pacar | *Sekar pacar* is the name of a *nitik* motif whose meaning consists of encouraging people to be *migunani*, or useful to others. *Pacar* (henna) flowers are very small, yellow in colour, form a fairly large shrub, and smell good. Hindus use henna flowers (*Impatiens balsamina* L.) widely in various rituals. |
| 6 | Arumdalu | *Sekar Arum Dalu* (*Cestrum nocturnum*) will bloom at night and spread its fragrance around. This means that as a human being, one should do good for others, especially for the surrounding community, regardless of the differences that exist. Like in the dark of the night (*nocturnum*), we still have to be wise about it. |
| 7 | Sekar Srigading | *Sekar sri gading*. *Sri gading* (*Nyctanthes arbor-tristis*) is a shrub that produces flowers that bloom after sunset. The flowers are white with red stems and produce a very fragrant smell when blooming. In the puppet story, Sri Batara Khresna took the time to plant *sri gading* on the border of the yards of the residences of his two wives, Dewi Rukmini and Dewi Sathyabama. Once they both smelled the fragrance of the sri gading flowers, the two, who did not have a harmonious relationship, became more tolerant of each other in their daily lives. Thus, *sri gading* flowers are considered a symbol of harmony. So, the *sri gading* motif can be interpreted as hope of living in peace and harmony. |

*3.3. Experiments*

In this section, the present researchers tested Batik Nitik 960 to provide evidence through conventional classification algorithms, including Multi Texton Co-Occurrence Descriptor (MTCD) [29], Image representation using Complete Multi-Texton Histogram (CMTH) [20], and Gray Level Co-occurrence Matrix (GLCM) [29] as features extraction and K Nearest Neighbor (KNN), Support Vector Machine (SVM), and Decision Tree (DT) as classifiers. In addition, the feature extraction time was calculated for a single image in seconds under the three feature extraction scenarios, as follows:

FE1: MTCD using six texton.
FE2: CMTH using 11 texton.
FE3: GLCM using four features (energy, entropy, contrast, and correlation).

The dataset was further divided into three parts: training, validation, and testing with 12, two, and two images, respectively. Meanwhile, performance was assessed by using accuracy calculated from the amount of data which was correctly predicted divided by the actual test data. Accuracy calculation is illustrated in Equation (1), using true positives (*TP*), true negatives (*TN*), false negatives (*FN*), and false positives (*FP*). The comparison results are presented in Table 3.

$$Accuracy = \frac{TP + TN}{TP + FN + FP + FN} \tag{1}$$

**Table 3.** Comparison results using three scenarios and three common classifiers.

| Feature Extraction | Accuracy | | | | | | | Time (s) |
| | KNN | | | | | SVM | DT | |
| | 1 | 3 | 5 | 7 | 9 | | | |
|---|---|---|---|---|---|---|---|---|
| FE1 | 0.51 | 0.53 | 0.49 | 0.50 | 0.49 | 0.71 | 0.69 | 0.0262 |
| FE2 | 0.50 | 0.51 | 0.48 | 0.50 | 0.48 | 0.67 | 0.68 | 0.0300 |
| FE3 | 0.49 | 0.49 | 0.48 | 0.5 | 0.48 | 0.47 | 0.58 | 0.0159 |

## 4. Conclusions

The researchers in this study collected Batik Nitik data from the Winotosastro pieces at PPBI Sekar Jagad in Yogyakarta, Indonesia. The photos of the batik were then handed over to batik experts for further identification of the motif name and philosophical value of each Batik Nitik motif. The next phase entailed image preprocessing and augmentation, generating the images which make up the dataset. The Nitik 960 dataset is a balanced dataset, as each class contains 16 images. The Batik Nitik 960 dataset is the first primary batik dataset containing a name and philosophical meaning for each motif and being validated by batik experts directly. This dataset is open access and aims to support research in batik classification, image retrieval, and batik generation. The result of this study (the existence of Batik Nitik 960 dataset) is expected to encourage innovation and afford solutions for future research of batik.

**Author Contributions:** A.E.M.: Writing, Methodology, Conceptualization; I.S.: Review & supervision; H.A.N.: Review & supervision. All authors have read and agreed to the published version of the manuscript.

**Funding:** Funding this work have been supported by the Center for Education Financial Services (Puslapdik) and Indonesia Endowment Funds for Education (LPDP) Number 03412/J5.2.3./BPI.06/10/2022.

**Institutional Review Board Statement:** Not applicable.

**Informed Consent Statement:** Informed consent was obtained from all subjects involved in the study.

**Data Availability Statement:** The dataset presented in this study is openly available at http://doi.org/10.17632/sgh484jxzy.3 (accessed on 23 January 2023).

**Acknowledgments:** The authors are very grateful to our university, PPBI Sekar Jagad Yogyakarta, and Syaifuddin for assistance in data collection.

**Conflicts of Interest:** The authors declare no conflict of interest.

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
