# Peer review of "Batik Nitik 960 Dataset for Classification, Retrieval, and Generator"

_data, 2023_

Round 1

Reviewer 1 Report

The paper is about a texture image dataset acquired from Batik motifs. As far as I understand, it is traditional to Indonesia. In this aspect, this dataset is significant to the community.

The authors mention that this is the first dataset to the public research. However, authors have previous studies about the topic. This should be justified.

Overall, the quality of the paper is good, except the level of English. Some example phrases are to be corrected, for instance in page 4, line 91 "...it takes 60 times the cropping process" -> "...cropping process requires 60 iterations.."

Also, the authors only present the image sets. It is not verified that the dataset is available for possible detection, recognition or classification tasks. My advice is to present at least two conventional methods' classification performance so that it forms a verifiable ground truth on the dataset.   

I can summarize my comments as follows:

1) The originality of the dataset should be justified

2) Level of English should be improved

3) Conventional classification algorithms should be tested on the dataset to form a ground truth for the researchers

Overall, the study presented in the paper is interesting.

Author Response

Point 1: The originality of the dataset should be justified

Response 1:

Thank you for your valuable feedback. We added a comparison between Batik 300 and batik Nitik 960 (lines 58 – 69).

Batik 300 is a general collection of 50 classes of batik motifs, while Batik 960 focuses on 60 classes of Nitik motifs, so Batik 960 significantly differs from Batik 300. Experts in the Indonesian batik community (Paguyuban Pecinta Batik Indonesia (PPBI) Sekar Jagad Yogyakarta and Universitas Muhammadiyah Malang) curated the motifs on the batik cloth. This curation guarantees the dataset validity and makes it possible to provide additional metadata in the form of motif names and motif philosophies unavailable in Batik 300.

Point 2: Level of English should be improved

Response 2: Thank you for your valuable feedback. We improve our English level with a professional proofreader. The manuscript provides track changes.

Point 3: Conventional classification algorithms should be tested on the dataset to form a ground truth for the researchers

Response 3: Thank you for your valuable feedback. We added the experiments section (lines: 141-160) as a ground truth for researchers. The Batik Nitik 960 dataset was tested using three-feature extraction method (MTCD, CMTH, GLCM) and three common classifiers (KNN, SVM, Decision Tree).

Feature Extraction Accuracy Time (s)
KNN SVM DT
1 3 5 7 9
FE1 0,51 0,53 0,49 0,50 0,49 0,71 0,69 0,0262
FE2 0,50 0,51 0,48 0,50 0,48 0,67 0,68 0,0300
FE3 0,49 0,49 0,48 0,5 0,48 0,47 0,58 0,0159

Reviewer 2 Report

The article introduced the Batik Nitik 960 dataset that was performed using patterns extracted from a piece of fabric with sixty Nitik patterns. Each of the 60 categories in the collection has 16 photographs, totally of 960 images (jpg format); each category contains four motifs rotated by 90, 180, and 270 degrees. Novel dataset can be used the training and evaluation of machine learning models for classifying, retrieving, etc. The authors mentioned that their dataset is the first Batik Nitik dataset that is freely accessible.

Comment: The authors should present detailed justification of the advantages of their novel dataset in comparison with existed Batik 300 dataset. The authors only wrote Batik 300 dataset “does not include adequate metadata to support meaningful analysis and interpretation. The Batik Nitik 960 dataset, on the other hand, is the first publicly available dataset that provides rich metadata, including the name of the motif, its category, and its philosophical meaning” (lines: 47-52). Please discuss how presented by authors metadata can help in the training and evaluation of machine learning models in comparison with other dataset.

Author Response

Point 1: The authors should present detailed justification of the advantages of their novel dataset in comparison with existed Batik 300 dataset. The authors only wrote Batik 300 dataset “does not include adequate metadata to support meaningful analysis and interpretation. The Batik Nitik 960 dataset, on the other hand, is the first publicly available dataset that provides rich metadata, including the name of the motif, its category, and its philosophical meaning” (lines: 47-52). Please discuss how presented by authors metadata can help in the training and evaluation of machine learning models in comparison with other dataset.

Response 1: Thank you for your valuable feedback. We added an explanation and comparison between Batik 300 and Batik Nitik 960 (lines 58-69). While “adequate metadata” may be ambiguous, we believe that “complete and comprehensive metadata” can open up the possibility of applying better techniques and algorithms to train and evaluate machine learning models. Unlike Batik 300, which only has a label, Batik 960 is equipped with motif names and sufficient explanations of philosophical meanings. This metadata, which is expertly validated, is a valuable characteristic that is used as additional information and features to perform a more accurate classification. Additionally, in practical implementation as search tools, the textual information in the metadata can be used for text-based queries, complementing the main aim of image-based queries.

Round 2

Reviewer 1 Report

The paper is significantly improved after the revision. The authors did sufficient improvements and necessary justifications are added. 

The paper will attract interest of the researchers in the computer vision fields related to texture based classification studies as well.

Author Response

Thank you for your valuable comments